# Enhanced Treatment of Pharmaceutical Wastewater by an Improved A²/O Process with Ozone Mixed Municipal Wastewater

**Jian Wang [1,2], Cong Du [2,3], Feng Qian [2,3,*], Yonghui Song [2,*] and Liancheng Xiang [2,3]**

[1] State Key Joint Laboratory of Environment Simulation and Pollution Control, School of Environment, Tsinghua University, Beijing 100084, China; jian-wan14@mails.tsinghua.edu.cn

[2] State Key Laboratory of Environmental Criteria and Risk Assessment, Chinese Research Academy of Environmental Sciences, Beijing 100012, China; ducongducong@126.com (C.D.); xianglc@craes.org.cn (L.X.)

[3] Department of Urban Water Environmental Research, Chinese Research Academy of Environmental Sciences, Beijing 100012, China

[*] Correspondence: qianfeng@creas.org.cn (F.Q.); songyh@craes.org.cn (Y.S.); Tel.: +86-10-8492-4787 (Y.S.)

**Abstract:** A pilot-scale experiment is carried out for treating mixed wastewater containing pharmaceutical wastewater (PW) and domestic wastewater (DW), by a process that is a combination of hydrolysis acidification-ozone-modified anaerobic–anoxic–aerobic-ozone (A²/O) (pre-ozone) or hydrolysis acidification-modified A²/O-ozone (post-ozone). The effects of different mixing ratios of PW and DW and pre-ozone treatment or post-ozone treatment on the removal of nitrogen and phosphorus and chemical oxygen demand (COD) are compared and studied. The optimal ratio of PW in mixing wastewater is 30%, which has the optimal COD removal efficiency and minimum biotoxicity to biological treatment. The pre-ozone treatment shows more advantages in removing nitrogen and phosphate but the post-ozone treatment shows more advantages in COD removal. Analysis of dissolved organic matter (DOM) demonstrates that post-ozone treatment has a more significant effect on the removal of fulvic acid and humic acid than the effect from the pre-ozone treatment, so the COD removal is better. Overall DOM degradation efficiency by post-ozone treatment is 55%, which is much higher than the pre-ozone treatment efficiency of 38%. Microbial community analysis reveals that the genus *Thauera* and the genus *Parasegetibacter* take great responsibility for the degradation of phenolics in this process. All the results show that the post-ozone treatment is more efficient for the mixed wastewater treatment in refractory organics removal.

**Keywords:** pharmaceutical wastewater; $O_3$; A²/O; DOM

## 1. Introduction

It has always been difficult to treat pharmaceutical wastewater (PW) to reach the effluent water quality standard, due to its characteristics of large differences in water quality and volume, complex composition, high pollutant content and poor biodegradability [1,2]. If biological treatment is carried out directly, the toxic substances in PW may affect the biological system, thus affecting the treatment effect [3,4]; many substances are heterocyclic organics which are difficult to eliminate by a biological treatment. As a result, these toxic residues become micropollutants and finally enter into the environment. Hence, some advanced treatment methods are needed to remove these micropollutants before they finally enter the waterbody [5].

Many oxidation methods have been employed for destroying toxic organic pollutants, due to their capacity to destroy almost any organic contaminant [6]. Among those methods, ozone ($O_3$) is often used to remove color and a variety of complex pollutants in water, due to its high oxidation

capacity [7–11]. The structure of organics can be destroyed by direct oxidation of ozone or by the indirect reaction of the hydroxyl radical produced by ozonation [12]. Most of the pharmaceuticals in PW could be completely destroyed by a small dosage of ozone ranging from 5 to 15 mg/L [8]. However, in most real water treatment, applied ozone was only partially sufficient to reduce some of pharmaceuticals due to the interference from other organics [13]. In spite of this, the implementation of ozone treatment was conducive to reducing the difficulty of the subsequent biological treatment by reducing pharmacological activity and toxicity.

Utilizing ozone to treat the effluent from biologically treated PW (post-ozone treatment) appears to be another effective means of increasing ozonation efficiency. Most of the organics are degraded by biological treatment, but non-biodegradable complex organic compounds still remain, which can only be degraded and detoxified by ozone. Therefore, ozone will make the ozonation more efficient and will eliminate the interference from other organics. When PW and domestic wastewater (DW) are mixed, the concentration of refractory organics from PW will be reduced, and the toxicity will be greatly reduced. As a result, more organics from PW will be more easily biodegraded. The biodegradability and toxicity of PW on biological treatment can be solved by mixing with DW. The impact of PW on biological processes was decreased due to the reduced concentration of toxic substances. However, some other studies suggested that pre-ozone treatment of PW before the biological treatment can effectively improve the biodegradability. Some refractory organics can be oxidized by ozone and broken down into biodegradable organics so as to reduce the whole toxicity of PW. Following this pretreatment, the result will be easily treated after mixed with DW and the partial ozone pretreatment by-products can be treated with only a fraction of the biodegradables which can be further degraded, thereby improving the overall removal rate of organic matter [13–16]. Therefore, due to the advantages and disadvantages of pre-ozone and post-ozone treatment, appropriate research is required to compare the effects of these two ozone treatment types on PW treatment.

In this study, an ozone pre-treatment process and an ozone post-treatment process combined with a novel improved anaerobic–anoxic–aerobic-ozone ($A^2$/O) process [17] were designed to treat PW, respectively. A pilot-scale experimental study was carried out on the mixed water of PW and DW by a combination of hydrolysis acidification-ozone-modified $A^2$/O (pre-ozone) or hydrolysis acidification-modified $A^2$/O-ozone (post-ozone). The improved $A^2$/O process is based on the $A^2$/O, which absorbs the advantages of Modified University of Cape Town (MUCT) and oxidation ditch processes, and develops a low-energy denitrification and dephosphorization process [18–20]. The anaerobic–aerobic combination process ($A^2$/O) is currently the main process for the treatment of high-concentration organic wastewater. In view of the difficulty in degradation of PW, the $A^2$/O process is chosen to enhance the removal of nitrogen and phosphorus and the DW and $O_3$ was used to reduce toxicity of PW and improve the biodegradability. The mixing ratio of PW and DW was controlled and adjusted to study the quality characteristics of the effluent under different working conditions. The dissolved organic matter (DOM) characteristics of the treated effluent were analyzed by three-dimensional fluorescence to compare the effects of different sections on PW treatment. The aim of this study was to (i) develop an efficient process utilizing $O_3$ to treat PW and (ii) compare the effects of pre-ozone and post-ozone on PW treatment.

## 2. Material and Methods

### 2.1. PW (Pharmaceutical Wastewater) and DW (Domestic Wastewater) Characteristics

The PW was collected from the effluence of a pharmaceutical wastewater treatment plant in Shenyang, China, and the DW was collected from a municipal sewage treatment plant in Shenyang, China. The PW of this plant belonged to a comprehensive wastewater composed of chemical synthetic wastewater and biological wastewater. The basic water quality of PW and DW is shown in Table 1. The PW is characterized by high levels of refractory organics and very low biodegradability.

**Table 1.** The basic water quality of pharmaceutical wastewater and domestic wastewater.

| Index | PW | DW |
|---|---|---|
| COD | 650 ± 150 mg/L | 300 ± 100 mg/L |
| BOD/COD | <0.3 | 0.4 ± 0.1 |
| TN | 45 ± 15 mg/L | 41 ± 9 mg/L |
| $NH_4^+$-N | 30 ± 2 mg/L | 31 ± 2 mg/L |
| TP | 3.0 ± 0.5 mg/L | 2.3 ± 0.3 mg/L |
| pH | 8.0 ± 0.5 | 7.5 ± 0.4 |

The main organic compounds in PW are listed in Table 2. A total of 222 organic compounds, including hydrocarbons, phenols, esters, ketoaldehydes and other heterocyclic compounds, were detected.

**Table 2.** The content of various organic compounds in PW.

| Category | Hydrocarbon | Phenols | Esters | Aldehydes and Ketones | Others |
|---|---|---|---|---|---|
| Species | 44 | 20 | 39 | 30 | 89 |
| Content | 22.5 ± 13.5% | 16 ± 5% | 29 ± 21% | 22.5 ± 14% | 29.5 ± 11.5% |

## 2.2. Improved $A^2$/O Process with Pre-Ozone and Post-Ozone Treatment

The main part of the improved $A^2$/O reactor was made of stainless steel. The effective volume was 3 m$^3$ and the volume ratio of each compartment was pre-anoxic pool: anaerobic pool: anoxic pool = 0.8:1:1; anoxic pool: aerobic pool = 1:3. A physical photograph of the improved $A^2$/O reactor is shown in Figure 1.

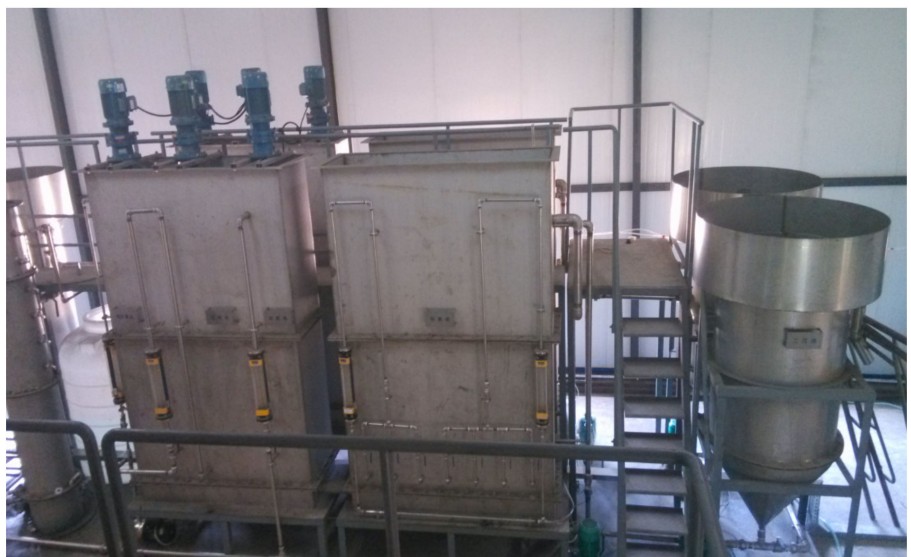

**Figure 1.** A physical photograph of the improved anaerobic–anoxic–aerobic-ozone$A^2$/O reactor.

Mechanical agitation was installed at the top of the pre-anoxic, anaerobic and anoxic zones. The bottom of the aerobic tank was provided with a perforated tube for aeration. The detailed operating parameters of the improved $A^2$/O reactor are shown in Table 3. The inoculated sludge of the reactor was taken from the sludge return channel of the sewage treatment plant in Shenyang.

**Table 3.** The improved anaerobic–anoxic–aerobic-ozone ($A^2$/O) operating conditions.

| Inlet Flow /(L/h) | Mixed Liquor Suspended Solid/(g/L) | Dissolved Oxygen/(mg/L) | Nitrification Liquid Reflux Ratio | Sludge Reflux Ratio | Hydraulic Retention Time/h | Sludge Residence Time/d |
|---|---|---|---|---|---|---|
| 200 | $3.26 \pm 0.74$ | $3.2 \pm 0.7$ | 100% | 50% | $14.3 \pm 0.8$ | $18 \pm 3$ |

Ozone treatment was divided into pre-treatment and post-treatment. For the pre-ozone treatment, the effluent of PW after hydrolysis acidification was directly entered into the ozone oxidation tank, and then mixed with different proportions of DW for follow-up biological treatment. The flow chart of the hydrolysis-acidification-ozone oxidation-improved $A^2$/O process is shown in Figure 2.

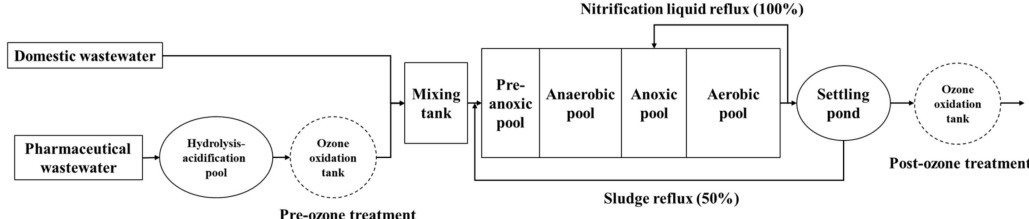

**Figure 2.** The flow chart of the hydrolysis–acidification–ozone oxidation-improved $A^2$/O process.

The experiment was divided into five stages according to the ratio of PW and DW, and the ratio increased gradually at a rate of 10%. The proportion of PW in mixed sewage was 10%, 20%, 30%, 40% and 50%, respectively. Each stage operated for thirty days and parameters did not change throughout the operation. The effective volume of the ozone oxidation tank was 0.24 $m^3$ with a hydraulic retention time (HRT) of 1 h. The flow rate of the ozone generator was 6 L/min. For the post-ozone treatment, the effluent from the improved $A^2$/O process directly entered the ozone oxidation tank. The dosage of ozone was 5 mg/L, 10 mg/L, 20 mg/L, and 30 mg/L, respectively. The hydrolysis–acidification and improved $A^2$/O process maintained the original working conditions. The proportion of PW in the mixing tank was 10%, 30%. and 50%, respectively. Samples were taken every 6 min after 25 min of the reaction.

*2.3. Three-Dimensional Fluorescence, Water Quality and Microbial Analysis*

Two sludge samples (pre-S and post-S) were taken from the settling pond of the pre-ozone process and the post-ozone process at the end of the experiment. The sludge samples were centrifuged at 10,000 rpm and the solid left was stored at −20 °C for microbial analysis. Liquid supernatant was filtered by 0.45 μm filters before chemical oxygen demand (COD), total organic carbon (TOC), total phosphorus (TP), total nitrogen (TN) and dissolved organic matter (DOM) analysis. All the analyses were conducted according to procedures which referred to the standard method. COD was determined by fast digestion spectrophotometry (HACH, New York, NY, USA), TOC was determined by a TOC tester (Shimadzu, Kyushu, Japan), $NH_4$-N was determined by the salicylic acid method, TN was determined by the persulfate oxidation method, and TP was determined by the digestion-ascorbic acid method. The DOM was analyzed by three-dimensional fluorescence excitation–emission matrix (EEM) spectroscopy (FP-6500, Jasco, Tokyo, Japan). The DNA of the sludge was taken by a soil DNA kit (OMEGA, Houston, TX, USA). A primer pair of 338F (5'- ACTCCTACGGGAGGCAGCAG-3')-806R (5'-GGACTACHVGGGTWTCTAAT-3') was selected for amplifying the V3–V4 region of bacteria.

## 3. Results and Discussion

*3.1. COD and N/P Removal by Pre-Ozone Treatment with Different Mixing Ratios of PW and DW*

Changes of COD, TN, TP and $NH_4^+$-N were monitored to investigate the effect of pre-ozone on PW treatment, as shown in Figure 3. The removal efficiency of COD was nearly consistent during

the five stages (different mixing ratios of PW and DW) except for the fourth stage due to the two weeks of instrument overhaul, shown in Figure 3a. The COD removal rate was stabilized in the later fourth stage when the instrument resumed normal operation. 70–80% of COD can be removed and COD in the effluent was about 80–120 mg/L in this system, which met the discharge standard of PW. The average COD removal efficiency in different stages could reach 77.6% (containing 10% PW, stage 1), 73.4% (containing 20% PW, stage 2), 65.1% (containing 30% PW, stage 3), 71.7% (containing 40% PW, stage 4) and 67.3% (containing 50% PW, stage 5), respectively. It showed that the COD removal efficiency was slightly decreased as the proportion of PW increased in the mixed wastewater, although the influent COD was not changed at the first three stages. This indicated that although pre-ozone oxidation was performed, there were still some refractory organic matters left in the mixing wastewater, which could not be fully treated by biological treatment, thereby resulting in an increase of COD in the effluent. The effluent COD began to increase significantly when the mixing ratio of PW was 40%, which indicated that the biological system was overloaded. The optimal mixing ratio of PW was 30%, with a COD removal rate of 72.1% and a low effluent COD of 97.2 mg/L. In contrast, the COD removal rate of PW can only reach 43% ($O_3$ oxidation) and 62% ($O_3$ + BAF) in HE's research [21]. Obviously, the COD removal is more effective in our research.

The nitrogen removal can be seen from Figure 3b,c. The reactor had been steadily operating for more than 200 days and the nitrogen concentration of PW was similar to DW, so there was not a significant effect on nitrogen removal of the system by PW addition. During the first three stages, the $NH_4^+$-N concentration of influent was about 19–29 mg/L, and the effluent was below 5 mg/L. Generally, the nitrogen removal efficiency was about 85%, which illustrated that the system had favorable denitrification ability due to the sufficient carbon source and oxygen. According to the internal reflux, the theoretical nitrogen removal efficiency was 71.4%. The average nitrogen removal efficiency for the first three stages were 75.14%, 75.56%, 74.23%, respectively, which were all above the theoretical value. This meant that nitrification and denitrification or endogenous respiratory denitrification both existed in the reactor, so nitrogen could be removed efficiently, although PW flowed into the system. The nitrogen concentration in the effluent was 5–10 mg/L.

As shown in Figure 3d, the phosphates concentration of influent was relatively low and had been maintained at 1.5–4.0 mg/L during the operation. Based on the microbial demand for nutrients, C:N:P = 100:5:1, the phosphates content of the system was not enough to maintain the microbial growth, hence, the phosphates removal efficiency was high. In the first three stages, the removal efficiency was all above 90%. At the fourth stage, polyphosphate accumulating organisms were inhibited due to an aeration equipment fault and the amount of sludge discharge was decreased, so the phosphates concentration of the effluent increased up to 1.3 mg/L. When the dissolved oxygen of the system recovered to above 2 mg/L, the phosphates removal capacity was restored.

After the pre-ozone treatment, the biotoxicity and biodegradability of PW were reduced and finally a good biological treatment efficiency was obtained. Assuming that the average removal rate of COD in DW was 85%, at least 39% COD in PW was degraded when the PW accounted for 30% (stage 3). It indicated that some refractory organic compounds were removed by co-metabolism of PW and DW. However, as the mixing ratio of PW increased further, the decrease of biochemical oxygen demand (BOD) led to a deficiency of nutrients in the reactor and an increase of biotoxicity, which led to a decline of COD removal efficiency. Hence, it showed that although PW was treated with ozone, the proportion of PW could not be too high. The optimal mixing ratio of PW was 30%, which has the optimal COD removal efficiency.

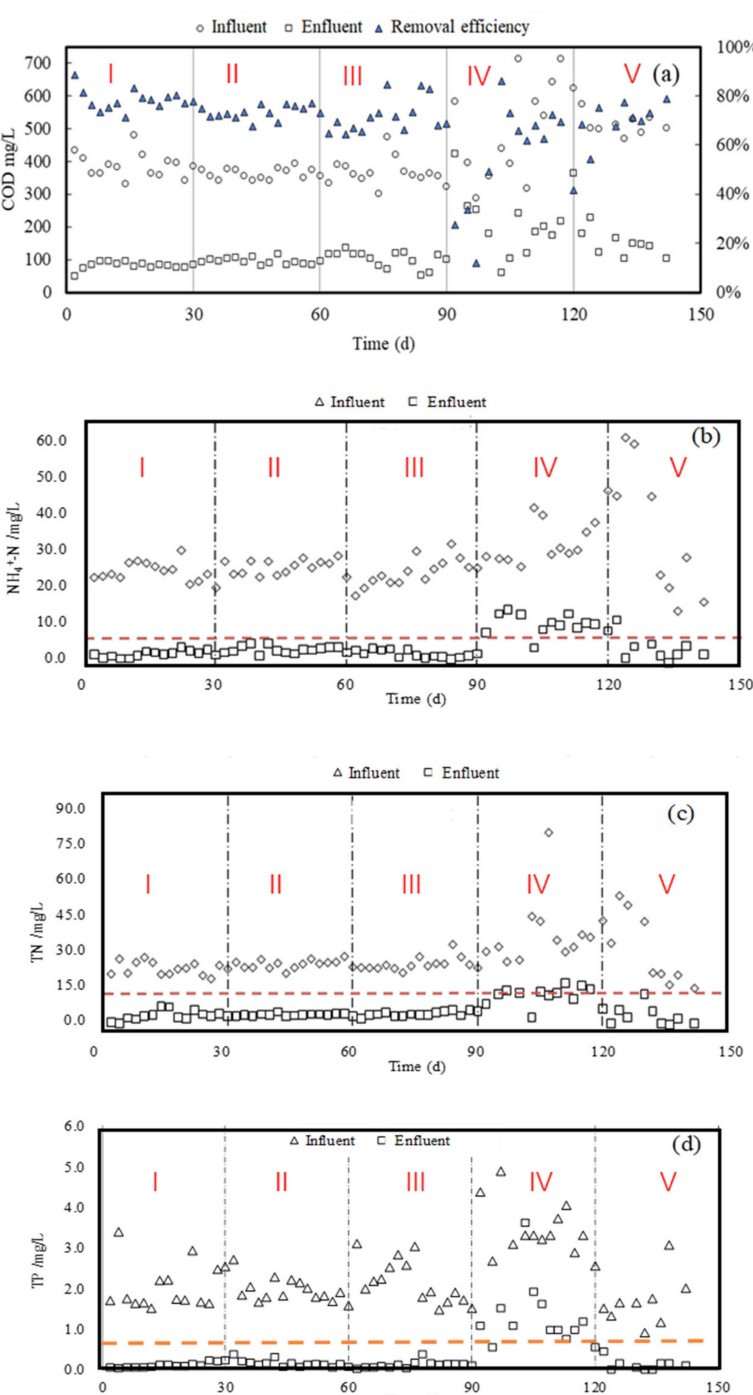

**Figure 3.** The variation of major water quality parameters in five stages with different mixing ratios of PW and DW: (**a**) chemical oxygen demand (COD), (**b**) $NH_4^+$-N, (**c**) total nitrogen (TN), and (**d**) total phosphorus (TP).

### 3.2. COD and N/P Removal by Post-Ozone Treatment after Biological Treatment

For post-ozone treatment, the mixed wastewater from the settling pond after biological treatment directly flowed into the ozone oxidation tank. Removal of the COD, $NH_4^+$-N and TP in the effluent during ozonation is shown in Figure 4. The effluent COD concentration increases with an increase of the PW component in the mixed wastewater, as shown in Figure 4a–c. Therefore, when the PW accounted for 50% of the wastewater, the effluent COD concentration reached 135.2 mg/L, while when PW accounted for 10%, the effluent COD was only 82.4 mg/L. This meant that more ozone was needed

to remove organic matter from the effluent when the PW level was high. For the effluent with 10% PW, the highest COD removal efficiency of 16.5% was obtained when the ozone concentration was 15 mg/L and the reaction time was 43 min. For the effluent with 30% PW, the highest COD removal efficiency of 24.4% was obtained when the ozone concentration was 20 mg/L and the reaction time was 37 min. For the effluent with 50% PW, the highest COD removal efficiency of 19.9% was obtained when the ozone concentration was 25 mg/L and the reaction time was 37 min. This showed that the ozone could effectively remove COD from the mixed effluent and some refractory organics were thoroughly mineralized. However, at the end of the reaction, the COD concentration increased slightly. This indicated that after ozone oxidation, some non-mineralized macromolecular refractory organics were degraded into small molecular organics, resulting in an increase in COD. At the same time, the effluent COD from another similar study about the PW by MBR without $O_3$ was always above 300 mg/L [22]. Therefore, it is necessary to carry out advanced treatment by ozone.

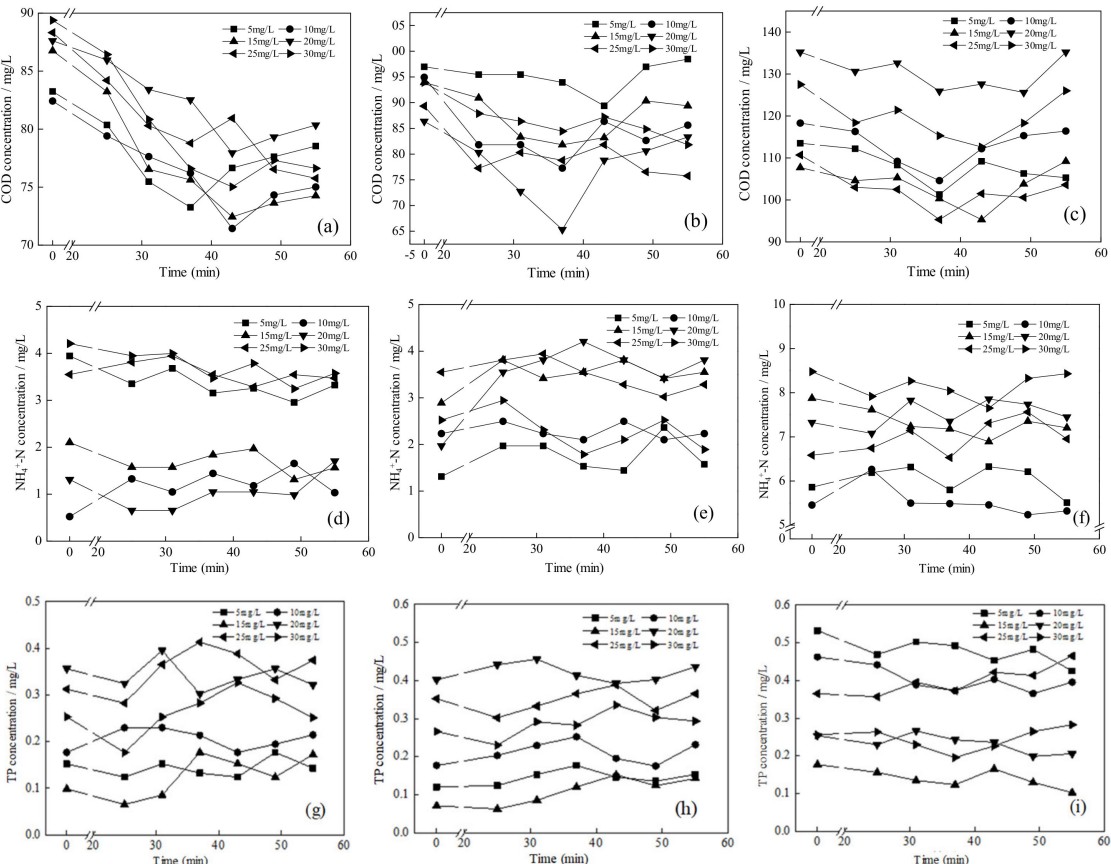

**Figure 4.** The variation of $NH_4^+$-N, TN and TP concentrations for post-ozone treatment: (**a,d,g**) 10% PW; (**b,e,h**) 30% PW; and (**c,f,i**) 50% PW.

The $NH_4^+$-N removal is shown in Figure 4d–f. For the influent with 10% or 30% PW, $NH_4^+$ concentration in the secondary sedimentation tank was below 5 mg/L, which met the discharge standard of pollutants for the municipal wastewater treatment plant. However, when the ratio of PW was 50%, the effluent $NH_4^+$-N could not meet the standard, which was 5–9 mg/L. This showed that the improved $A^2$/O process with post-ozone treatment could effectively degrade and eliminate $NH_4^+$-N in the mixing influent with low PW ratio. When the PW ratio increased up to 50%, the biological treatment efficiency was lowered, and excessive $NH_4^+$-N remained. However, during 43 min reaction, the $NH_4^+$ concentration of the influent nearly did not change compared to the effluent after ozone oxidation. This indicated that the ozone could hardly catalyze at the conversion rate of $NH_4^+$-N into $NO_3^-$ or $NO_2^-$ in this case. By contrast, ozone may be more involved in the oxidation of nitrite and organic

nitrogen. Hence, the ozone had little effect on $NH_4^+$ removal. Similar to the changes of $NH_4^+$-N, the phosphates removal efficiency is shown in Figure 4g–i. During the 43 min reaction, the variation of the phosphates concentration was inconspicuous after ozone oxidation, which indicated that the effect of ozone on phosphates removal was not significant. However, the phosphates concentrations of the secondary sedimentation tank effluent were all below 0.5 mg/L when the proportion of PW was 10%, 30% or 50%, which met the discharge standard of pollutants for the municipal wastewater treatment plant. This showed that the post-ozone treatment mainly had a significant reduction effect on COD in wastewater and its main role was to deeply treat refractory organic matter.

A pilot scale experiment was also carried out to analyze the treatment efficiency for pre-ozone and post-ozone treatment. When the mixing ratio of PW was 50%, the conventional water quality index of every treatment unit was measured as shown in Figure 5. Due to the difference in winter and summer climates, there was a certain difference in water quality between DW and PW. However, it did not affect the evaluation of the processing effect of each unit. It showed that the removal of COD, TN, $NH_4^+$-N and TP were not obvious before and after pre-ozone treatment, as shown in Figure 5a, indicating that the main role of ozone was to oxidize macromolecules into small molecules and the pollutants removal were mainly dependent on the subsequent $A^2/O$ processes. The following biological treatment obtained good results with the benefits of pre-ozone treatment. The overall removal rate of COD, TN, $NH_4^+$-N and TP reached 69%, 84%, 90% and 92%, respectively; these rates were higher than that of post-ozone treatment except for COD shown in Figure 5b. For post-ozone treatment, the overall COD removal rate reached 74%, which was 6.8% higher than the pre-ozone treatment. This indicated that the post-ozone treatment was more effective in achieving efficient removal of COD.

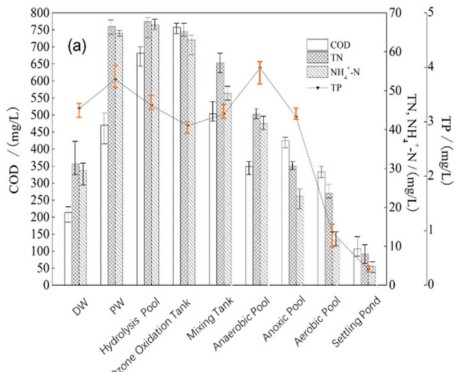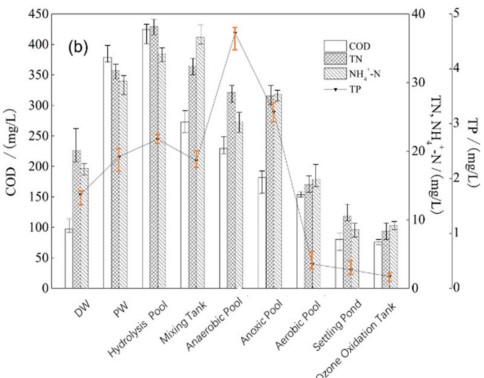

**Figure 5.** The variation of the COD, $NH_4^+$-N, TP and TN concentrations for (**a**) pre-ozone and (**b**) post-ozone treatment (50% PW). (Each data is from three parallel samples).

### 3.3. DOM Removal Characters for Pre-Ozone and Post-Ozone Treatment

In order to identify the removal characters of organic matter, especially refractory organic matter by ozone treatment, DOM derived from every unit of the system was analyzed by excitation emission matrix spectra and the regional integration method. The excitation emission matrix spectra of each process effluent containing 50% PW and 50% DW are shown in Figures 6 and 7. Five main components were classified based on different fluorescent regions which were aromatic protein I (I), aromatic protein II (II), fulvic acid (III), soluble microbial metabolite (IV) and humic acid (V). The regional standard integrals of five components are shown in Tables S1 and S2. The main components of DW and PW were fulvic acid III and aromatic protein II. The content of humic acid V and aromatic protein I in DW were relatively small. By contrast, these five components were all abundant in PW. Hence, the main sources of humic acid V and aromatic protein I were from PW.

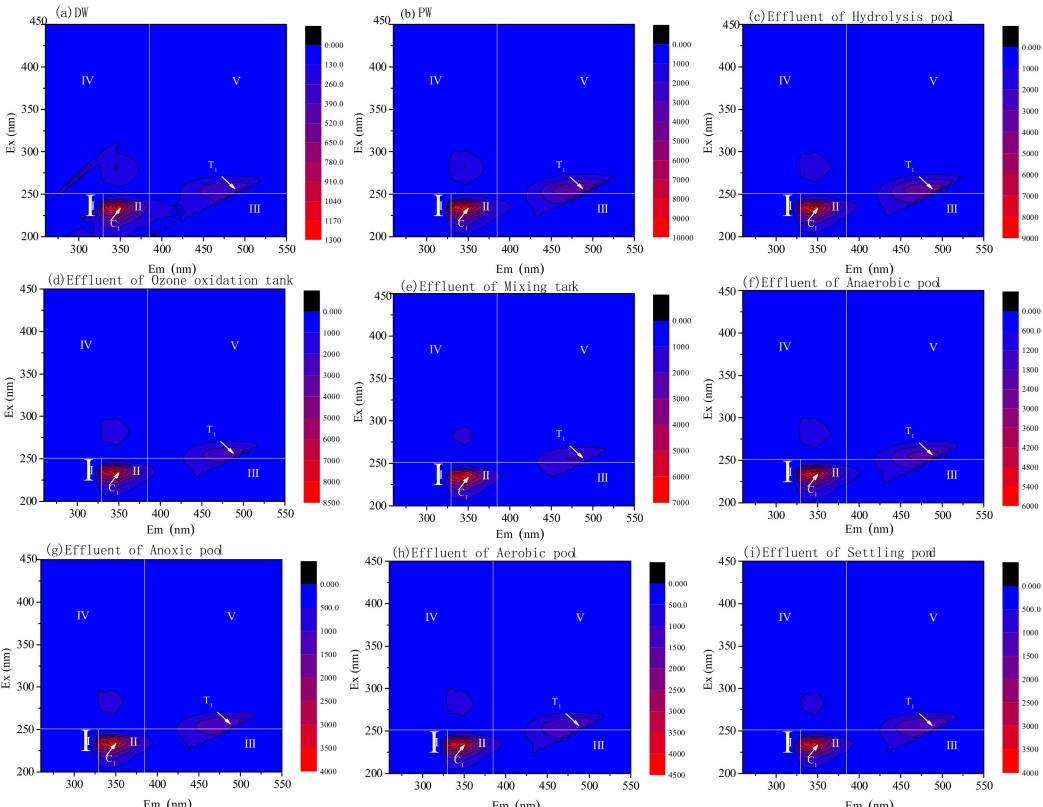

**Figure 6.** The excitation–emission matrix (EEM) spectra of each process effluent for pre-ozone (50% PW). ((**a**) DW sample; (**b**) PW sample; (**c**) Effluent of Hydrolysis pool sample; (**d**) Effluent of Ozone oxidation tank sample; (**e**) Effluent of Mixing tank sample; (**f**) Effluent of Anaerobic pool sample; (**g**) Effluent of Anoxic pool sample; (**h**) Effluent of Aerobic pool sample; (**i**) Effluent of Settling pool sample).

For pre-ozone treatment, ozone had a more obvious effect on the removal of fulvic acid and humic acid: nearly 19% of fulvic acid and 25% of humic acid were removed by ozonation. For other components, the removal effect was not significant, only 7% of aromatic protein I and 8% of aromatic protein II were removed. The content of soluble microbial metabolites increased slightly, probably due to the fact that hydrolysis bacteria flowed into the ozone instrument. Most organic components in the mixing wastewater were then gradually removed by the subsequent $A^2$/O process treatment. Finally, the total removal efficiency of aromatic protein I reached 45.3%, which was higher than that achieved for the other organic components. The least removal efficiency occurred for fulvic acid and humic acid, which reached 27.2% and 27.1%, respectively. This indicated that biological treatment was more significant for the removal of microbial metabolites and aromatic protein. Compared to pre-ozone treatment, the post-ozone treatment achieved better overall removal efficiency of humic acid and fulvic acid that treatment reached 87.9% and 73.9% efficiency, respectively. This was due to the high efficiency of ozone treatment in the last process of the treatment, which eliminated much of the organic matter in the effluent of biological treatment. After biological treatment, humic acid was decreased by 40.5%, while aromatic protein I only decreased by 14.1%. However, after treated by ozone oxidation, the concentration of humic acid decreased more than double to 87.9% compared to biological treatment. Nearly 80% of humic acid and 65% of fulvic acid were removed by ozone treatment alone, and the remaining components were also removed by about 25%. This indicated that when most of the organics were removed by biological treatment, ozone could remove the remaining refractory organics more efficiently. The organic components were highly removed by ozonation at the end of treatment. Finally, 55% of the DOM was removed by post-ozone treatment, whose removal rate was higher than pre-ozone treatment of 38%. It can be seen that post-ozone was more effective for degradation of pollutants.

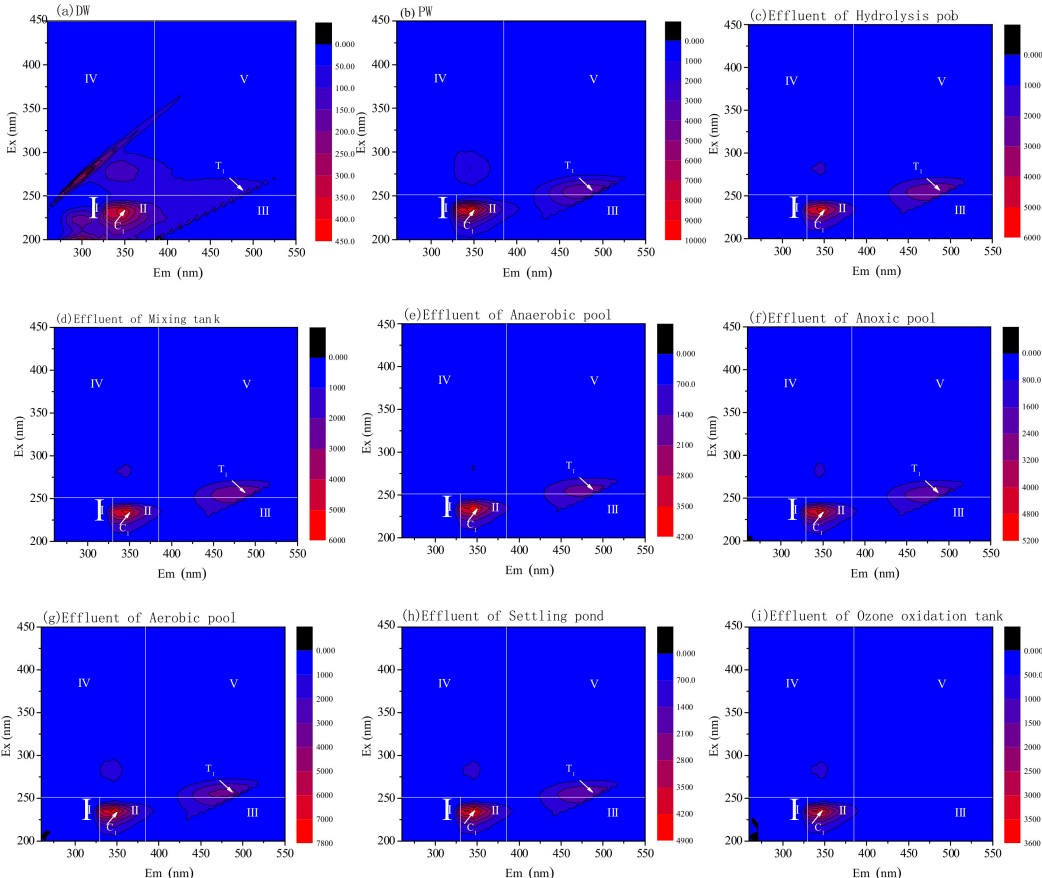

**Figure 7.** The excitation emission matrix spectra of each process effluent for post-ozone (50% PW). ((**a**) DW sample; (**b**) PW sample; (**c**) Effluent of Hydrolysis pool sample; (**d**) Effluent of Mixing tank sample; (**e**) Effluent of Anaerobic pool sample; (**f**) Effluent of Anoxic pool sample; (**g**) Effluent of Aerobic pool sample; (**h**) Effluent of Settling pool sample (**i**)) Effluent of Ozone oxidation tank sample).

### 3.4. Microbial Community Analysis

The microbial community of three biological pools in the modified A$^2$/O system was analyzed by high-throughput illumina sequencing to investigate the effects of pre-ozone and post-ozone impacts on the biological system, as shown in Figure 8. At the genus level, 344 genera were detected for pre-ozone treatment, among 21 genera with richness greater than 0.8%, accounting for 61.7–64.6%. of the total sequence, shown in Figure 8a. *Thauera* was the most abundant microbe whether in OX1 (anaerobic pool) or AN1 (anoxic pool) or HY1 (aerobic pool), accounting for 14.8%, 16.4% and 19.5%, respectively. The genus *Thauera* is a type of gram-negative bacteria, belonging to the family Betaproteobacteria. Most of them are rod-shaped and capable of denitrification and degrading aromatic pollutants, which extensively exist in diverse wastewater treatments [23,24]. The enrichment of *Thauera* in the biological treatment system might be attributed to the high concentration of phenol in PW. Another high content genus *Methyloversatilis* is a kind of gram-negative bacteria under the *Rhodocyclaceae* family, which accounted for 4.0~4.5% in biological system for pre-ozone treatment, and it was discovered for its ability to degrade phenolic compounds [25]. Both *Thauera* and *Methyloversatilis* belong to the *Rhodocyclaceae* family, which indicates that *Rhodocyclaceae* plays a significant role in the biodegradation of organic pollutants in pre-ozone treatment. Other genera, such as *Parasegetibacter*, *Acidovorax*, *Hydrogenophaga*, accounting for 3%, have been reported to have certain effects on denitrification and phosphorus removal [26–29].

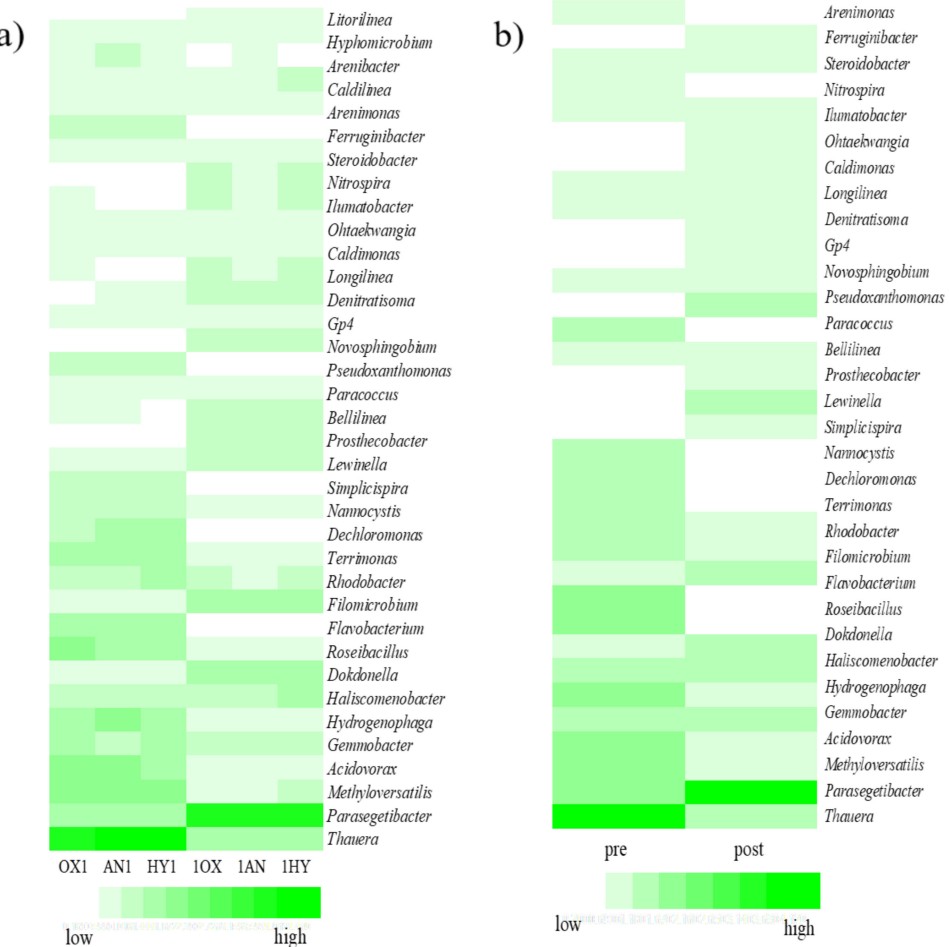

**Figure 8.** (**a**) The bacterial heatmap at phylum level in the anaerobic pool (OX), anoxic pool (AN) and aerobic pool (HY), containing 50% PW (pre-ozone: OX1, AN1, HYl, post-ozone: 1OX, 1AN, 1HY) and (**b**) microbial communities of high content PW in the pre-ozone treatment and post-ozone treatment process.

For the post-ozone system, 363 genera were observed with 23 genera more than 0.8%, accounting for 47.6–49.9% of the total sequences. Among them, *Parasegetibacter* was the predominant genus, accounting for 14.6%, 14.6% and 15.7% in 1OX (anaerobic pool), 1AN (anoxic pool) and 1HY (aerobic pool) respectively. *Parasegetibacter* is a kind of gram-negative bacteria belonged to the *Chitinophagaceae* family, which has been reported to be enriched in the process of treating tetrabromobisphenol A [30]. Compared to the pre-ozone treatment, the relative abundances of many genera were reduced, which indicated that the toxic effects of PW on microorganisms in biological treatment. In order to identify major functional microorganisms in PW treatment, the ratio of PW in the influent was gradually increased. With the increase of the ratio of the PW, microbial structure changed obviously, shown in Figure 8b. The proportion of *Sphingomonas* sp., *Pseudomonas aeruginasa*, *Brevibacterium* sp., *Pseudomonas aureofaciens*, *Sphingomonas pacuimobilis*, *Xanthomonas maltophilia*, *Sphingomonas pacuimobilis* and *Thauera* gradually increased, which indicated that these microorganisms might have great contribution to the degradation of refractory pollutants. The contents of phenolics such as p-methylphenol and phthalate esters such as dibutyl phthalate were relatively higher in PW. Therefore, the removal of these organic pollutants might be related to these bacteria. However, when 100% PW was flowed into the biological system, it failed to removal pollutants efficiently and massive microorganisms were dead. It turned out that the co-existence of DW and PW was critical to accelerate refractory organics degradation.

## 4. Conclusions

Both pre-ozone treatment and post-ozone treatment combined with an improved $A^2$/O process can efficiently degrade the nitrogen, phosphate and COD in mixing wastewater containing PW and DW. The pre-ozone treatment had more advantages in removing nitrogen and phosphate than the post-ozone treatment but in COD removal. The optimal mixing ratio of PW and DW in pre-ozone treatment was 30% with the total effluent nitrogen less than 10 mg/L, phosphorus less than 0.5 mg/L and COD less than 100 mg/L. For post-ozone treatment, the highest COD removal efficiency of 24.4% was obtained when the ozone concentration was 20 mg/L and the reaction time was 37 min. DOM analysis showed that post-ozone treatment was more efficient for the degradation of fulvic acid and humic acid and thereby received good removal efficiency of COD. Microbial community analysis revealed that the genus *Thauera* and the genus *Parasegetibacter* were dominant groups for pre-ozone and post-ozone treatments, respectively, which took great responsibility for the degradation of the refractory organics, especially for phenolics. Overall, the post-ozone treatment was more efficient for the mixed wastewater treatment in refractory organics removal.

**Supplementary Materials:** The following are available online at http://www.mdpi.com/2073-4441/12/10/2771/s1, Table S1: Regional integration of each process effluent for pre-ozone (50% PW), Table S2: Regional integration of each process effluent for post-ozone (50% PW).

**Author Contributions:** Conceptualization, J.W., C.D. and F.Q.; methodology, J.W. and C.D.; software, F.Q.; validation, Y.S. and L.X.; formal analysis, L.X.; investigation, J.W., C.D. and F.Q.; resources, Y.S.; data curation, J.W.; writing—original draft preparation, J.W.; writing—review and editing, F.Q.; visualization, J.W.; supervision, Y.S. and L.X.; project administration, Y.S.; funding acquisition, Y.S. All authors have read and agreed to the published version of the manuscript.

**Funding:** This research was funded by National Major Program of Science and Technology for Water Pollution Control and Governance (Fund number, 2012ZX07202-005, 2018ZX07601-003, PR China). Research Project on Comprehensive Programme and Management Platform of Environmental Protection for the Main Stream of the Yangtze River and Typical Cities (2019-LHYJ01-0104).

**Conflicts of Interest:** The authors declare no conflict of interest.

## Abbreviations

| | |
|---|---|
| $A^2$/O | anaerobic–anoxic–aerobic-ozone |
| COD | chemical oxygen demand |
| BOD | biochemical oxygen demand |
| PW | pharmaceutical wastewater |
| DW | domestic wastewater |
| DOM | dissolved organic matter |
| TP | total phosphorus |
| TN | total nitrogen |
| TOC | total organic carbon |
| HRT | hydraulic retention time |
| SRT | sudge residence time |
| DO | dissolved oxygen |
| MLSS | mixed liquor suspended solid |

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
