# Peer review of "Enhanced Treatment of Pharmaceutical Wastewater by an Improved A2/O Process with Ozone Mixed Municipal Wastewater"

_water, doi:10.3390/w12102771_

Round 1

Reviewer 1 Report

This paper deals with the pilot scale treatment of pharmaceutical water with ozone. The strength of the paper is that the application is assessed at larger scale than the laboratory scale. But the paper looks more like a report than an article with few scientific explanations. Nevertheless, it fits the scope of Water. I suggest major revision before publication:

  • The Supplementary Materials were not provided with the paper, so some parts are difficult to assess objectively.
  • The acronyms need to be define the first time they appear. For example, in the abstract, PW, DW, DOM…
  • I find that the Figure of the installation needs to be in the main text.
  • P8 line 274, I think it is the title of the next section?
  • Add more scientific explanations and comparison with other similar works.

Author Response

Dear reviewer,

Thank you for your kind comments, I have revised my paper carefully and responded all of your comments at the attachment. I am looking forward to hearing from you again.

Best Wishes

Reviewer 2 Report

In this manuscript, the application of a pilot-scale modified A2O process coupled with ozonation was conducted for treating pharmaceutical wastewater. It was compared performance evaluation of characterization of organic matter removal by applying advanced oxidation process depends on pre- or post ozonation. However, the results that are not an organized description needs to be modification regarding target water quality parameters. In that case of normal water quality parameters, the results revealed higher efficiency at the pre-ozonation process. But the post ozonation process was shown more efficient for treating refractory organics removal. So, I would recommend setting on specific objectives and novelties of this study. Please see the details on the comments as below:

  1. The title need to be modified more simple sentence.
  2. The full name need to describe at the first statement of abbreviation in abstracts (i.e. PW, DW)
  3. Intruduction mus be modified details description on the novelty of this manuscript comparing Pre- and Post- ozonation.
  4. What was the purpose of emphasis on the performace evalution on the mixing ratio of DW and PW in this study? 
  5. Why not measure suspended solid parameter on the feed?
  6. Schematic diagram of proposed process need to be include at the maintex.
  7. Figure 2; Why is show up the diferent levels of intinal concentration on COD, NH4, and TP with dosing rate of ozone. 
  8. Figure 3; Please replace figure as high resolution image and introduce standard deviation and numbers of samples. 

Minor comments

 All the table need to present average and standarad deviation of parameter.

 L.67 MUCT need to be describe full name. 

 L.85 Table.1 Please check the sub- and super- script.

 L.118 Manufacture information of TOC was missed. Pleas use term of analyzer instead of tester.

 L.126 Spell out of "Radio".

 Figure 1. should be mark stage number on the graph.

Author Response

Dear reviewer,

Thank you for your kind comments, I have revised my paper carefully and responded all of your comments at the attachments. I am looking forward to hearing from you again.

Best wishes

Jian WANG

Round 2

Reviewer 1 Report

Dear authors,

The comments were well-addressed, the paper can be accepted.

Best regards

Reviewer 2 Report

 I have reviewed the author's response. I would recommend accepting the current form. It was fully addressed the reviewer's comments.